# Niche construction mediates climate effects on recovery of tundra heathlands after extreme event

**Victoria T. González** [1,2]*, **Bente Lindgård**[1], **Rigmor Reiersen**[1], **Snorre B. Hagen**[2], **Kari Anne Bråthen**[1]

**1** Department of Arctic and Marine Biology, UiT – Arctic University og Norway, Tromsø, Norway,
**2** Department of Ecosystems in the Barents Region, NIBIO - Norwegian Institute of Bioeconomy Research, Svanvik, Norway

* victoria.gonzalez@nibio.no

**Data Availability Statement:** Data are available at https://hdl.handle.net/11250/2722169.

**Funding:** The study was funded by a grant from UiT- The Arctic University of Norway provided to

## Abstract

Climate change is expected to increase the frequency and intensity of extreme events in northern ecosystems. The outcome of these events across the landscape, might be mediated by species effects, such as niche construction, with likely consequences on vegetation resilience. To test this hypothesis, we simulated an extreme event by removing aboveground vegetation in tundra heathlands dominated by the allelopathic dwarf shrub *Empetrum nigrum*, a strong niche constructor. We tested the hypothesis under different climate regimes along a 200-km long gradient from oceanic to continental climate in Northern Norway. We studied the vegetation recovery process over ten years along the climatic gradient. The recovery of *E. nigrum* and subordinate species was low and flattened out after five years at all locations along the climatic gradient, causing low vegetation cover at the end of the study in extreme event plots. Natural seed recruitment was low at all sites, however, the addition of seeds from faster growing species did not promote vegetation recovery. A soil bioassay from 8 years after the vegetation was removed, suggested the allelopathic effect of *E. nigrum* was still present in the soil environment. Our results provide evidence of how a common niche constructor species can dramatically affect ecosystem recovery along a climatic gradient after extreme events in habitats where it is dominant. By its extremely slow regrowth and it preventing establishment of faster growing species, this study increases our knowledge on the possible outcomes when extreme events harm niche constructors in the tundra.

## Introduction

Extreme events are short-term changes in the climate or environment that can have long-term effects. Extreme events can be caused by for example climate change, such as winter warming episodes [1, 2], or increased herbivore disturbance [3] and human activity. These events are

Victoria T. Gonzalez. The publication charges for this article have been funded by a grant from the publication fund of UiT- The Arctic University of Norway. The funders had no role in study design, data collection and analysis, decision to publish, or preparation of the manuscript.

**Competing interests:** No authors have competing interests.

expected to increase in frequency and intensity in northern regions which can play an important role in shaping tundra ecosystems [4].

The impact of these extreme events can affect the recovery capacity of the dwarf shrub tundra which is usually dominated by a few shrub species [5]. Though most studies focus on effects and changes of deciduous tundra shrubs, there are still gaps in our understanding of how evergreen shrubs respond to climate change and disturbance [6]. For example some evergreen dwarf shrubs appear to be both sensitive [7, 8] and resistant [9] to climatic variation, hence it can be challenging to predict the consequences of extreme events in the tundra.

Furthermore, the outcome of extreme events might also be mediated by species interactions. For example, niche constructor species that modify their own or other species environments through their metabolism and legacy effects (*sensu* Mathews et al. 2014), may represent a structuring force after disturbance in tundra ecosystems [10], and may therefore modify the expected outcome of extreme events by favouring only those species that tolerate these structuring forces. Nevertheless, studies on the increase or decrease of evergreen shrubs are less common than those on deciduous shrubs and deeper knowledge of their effects on northern ecosystems is needed [6].

*E. nigrum* L. [11, 12] is an evergreen dwarf shrub which is common in tundra ecosystems [13]. *E. nigrum* has niche constructing capacities which are related to it being an allelopathic species [14]. It produces and releases large amounts of phytochemicals which are known to inhibit growth and establishment of other vascular plant species [15–17]. Furthermore, it forms dense mats of clonal growth, allowing only a few species to establish [18] and it has heavy recalcitrant litter [19] which impairs soil microbial processes [14].

There is evidence that niche construction by *E. nigrum* can mediate the recovery of heavy disturbed *E. nigrum* heathlands in continental areas, by restricting the growth of faster growing subordinate species [20–22]. Furthermore, *E. nigrum* has recently been found to be resistant to climatic fluctuation [9] and to mediate the positive effects of increasing summer temperatures in tundra plant communities [23]. Specifically, the expected increase in diversity and biomass of faster growing species associated with higher summer temperatures was shown to be less apparent when *E. nigrum* was present in the plant community, especially at high abundance. In contrast, dwarf shrub abundance was correlated with *E. nigrum* presence in plant communities [23–25]. In combination, this suggest that niche construction by *E. nigrum* might be the decisive factor determining the recovery of tundra heathlands after extreme events independent of climate. If so, no vegetation shifts are expected following extreme events even under favourable climates in *E. nigrum* dominated heathlands.

To test this hypothesis, we aimed to imitate the natural disturbance caused by extreme events at a sufficient level as to assess their effects on ecosystem recovery of *E. nigrum* heathlands under different climates. We conducted a manipulation experiment in a total of 144 plots in five *E. nigrum* dominated tundra heathlands located along a 200-km gradient, from continental to oceanic climate in Northern Norway with sharp contrasts of temperature and precipitation (Fig 1). We simulated an extreme event by removing all aboveground vegetation and studied the recovery of the tundra heathlands over 10 years.

Based on the lasting niche constructing abilities of *E. nigrum* (e.g. Aerts et al 2010) and the temperature mediating effect of *E. nigrum* presence in tundra plant communities [23], we hypothesize a slow recovery of all the disturbed plots along the entire climatic gradient. Furthermore, we expect the heathlands to have a low regenerative capacity through seed bank and seed rain [26], and that the addition of seeds from several faster growing species will not promote a faster recovery of the disturbed plots due to the allelopathic effect of *E. nigrum* on seed germination and seedling growth [16].

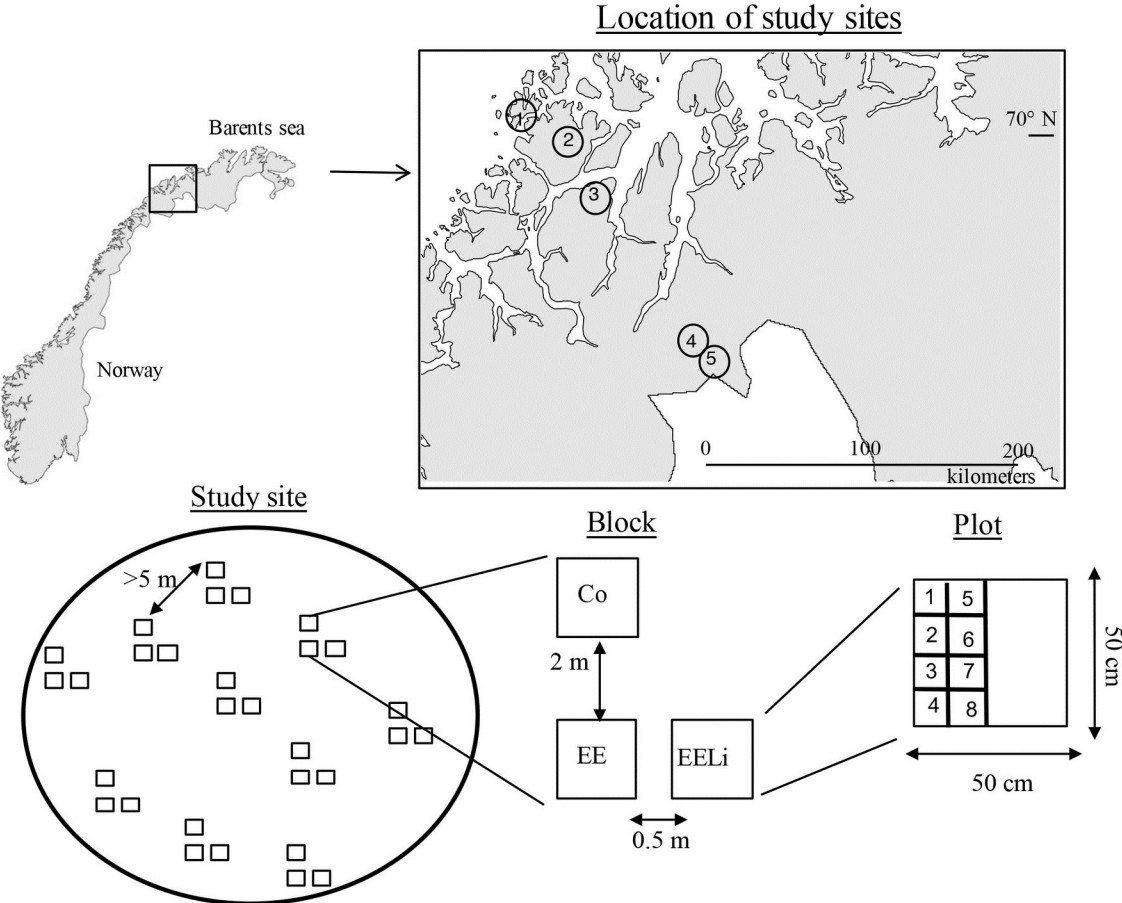

**Fig 1. Location of the study sites in Northern Norway.** 1) Rebbenes island; 2) Skogsfjord; 3) Snarby; 4) Skibotn and 5) Guhkesjávri. Each study site contained 10 blocks with three treatment plots each: (a) Control (Co): plot left untouched; (b) Extreme event (EE): all plant biomass was cut at ground level and discarded; (c) Extreme event + litter (EELi): all plant biomass was cut at ground level and left as a litter layer. The numbers on the plot correspond to the species in the seed addition experiment. See text for further details. For site characteristics see Table 1.

## Material and methods

### Study sites

Study sites were in areas with little to no easy access to the public on state owned land and therefore no permits were needed to establish and follow up the study. The study took place between 2009 and 2019 in five *E. nigrum* dominated tundra heathlands in northern Fennoscandia between 69.17–70.02˚N and 18.75–20.9˚ E (Fig 1). The study sites were above in the tree line ecotone along a 200-km climatic gradient, ranging from oceanic to continental conditions. Snow cover at the study sites remains until mid-June and bedrock is mainly gabbro (Table 1). The study sites were chosen according to the following criteria: they were situated just above the tree line ecotone, had a minimum of 90% cover of *E. nigrum*, and had a maximum slope of 5 degrees. The sites varied between exposed ridges to slightly sheltered depressions, but all had similar vegetation composition. Besides *E. nigrum*, other less abundant species were *Betula nana*, *Vaccinium uliginosum*, *Vaccinium myrtillus* and *Vaccinium vitis-idaea* [12]. The presence of herbivores inside the experimental plots was monitored through pellet counts in August each year. The pellet counts were minimal, averaging 4.5±1.05 plots

**Table 1. Characteristics of the study sites.**

|  | Rebbenes | Skogsfjord | Snarby | Skibotn | Guhkesjávri |
|---|---|---|---|---|---|
| Longitude/Latitude | 70˚ 10'; 18˚45' | 69˚ 57'; 19˚ 15' | 69˚ 45'; 19˚ 30' | 69˚ 14'; 20˚ 33' | 69˚ 10'; 20˚ 42' |
| Elevation (m a.s.l.) | 10 | 264 | 266 | 580 | 510 |
| Mean annual temp. (˚C) | 2.2 | 0.5 | 0.6 | -2 | -1.9 |
| Mean annual prec. (mm) | 871.18 | 860.49 | 917.19 | 634.16 | 628.2 |
| Mean summer temp. (˚C, June-July-August) | 12.37 | 11.77 | 12.04 | 11.96 | 11.64 |
| Mean summer prec. (mm, June-July-August) | 218.75 | 179.86 | 172.16 | 189.75 | 202.98 |
| Continentality index | 21.8 | 24.2 | 23 | 30.1 | 31 |
| Freezing days (mean/year) | 39 | 30 | 5 | 125 | 153 |
| Soil pH | 4 | 4 | 4.5 | 3.5 | 4 |
| Soil organic horizon depth (cm) | 5–10 | 3–10 | 2–5 | 3–10 | 5–10 |

Climatic variables were calculated based on data from the six study years using on-site soil surface temperature loggers, hence temperatures are measured at vegetation level. Precipitation data was gathered from publicly available databases (i.e. senorge.no).

with pellets per year out of 144 plots, and belonged to reindeer (*Rangifer tarandus tarandus*), small rodents such as Norwegian lemming (*Lemmus lemmus*), ptarmigan (*Lagopus lagopus*), grey-sided mountain hare (*Lepus timidus*) and in the island of Rebbenes (Fig 1) the greylag goose (*Anser anser*).

## Experimental design

We used a one-factor ANOVA design with extreme event (i.e. removal of all above ground vegetation) as the experimental treatment at each site (Fig 1). Removal of all above ground vegetation was meant to imitate heavy interacting extreme events such as winter warming episodes [7], burrowing by lemmings/voles [27, 28], or pathogens attacks [29] where large parts (if not all) of the aboveground vegetation dies or might be removed. Because certain types of extreme events can leave a litter layer (for example, a repeated winter warming event will kill the vegetation but not remove it, while voles cut the vegetation at ground level and remove it), this treatment had two sub-treatments: the cut vegetation was left as a litter layer or the cut vegetation was discarded, leaving a bare plot.

In June 2009, ten blocks were established in each of the sites (Fig 1). Blocks were placed a minimum of five meters apart to avoid including the same *E. nigrum* individual in different blocks, since long-lived, clonally reproducing species can have multiple modular units [30]. Unfortunately, two blocks were removed from the study due to unexpected field work problems (one at Skibotn and one at Guhkesjávri, locations 4 and 5 respectively, Fig 1), leaving 48 blocks and 144 plots in total. Each block included three treatment plots of 50 cm x 50 cm each, i.e. control, extreme event without litter and extreme event with litter (Fig 1). The two extreme event plots were assigned treatments in a random manner and placed beside each other with a 50-cm separation and approximately 2 meters downslope of the control plot (Fig 1). Vegetation was cut with standard garden shears at ground level including the vegetation in a 20-cm buffer zone between plots. Furthermore, all plots (except control plots) were trenched around with a spade which was 30 cm long to eliminate any below-ground effects of the vegetation adjacent to the plot.

## Environmental measurements

Three temperature loggers (Thermochron iButtons®) were placed at ground level at each site to measure soil surface temperature every 3 hours all year round (during 2010–2019). This

allowed us to account for microclimatic variations at site level, e.g. temperature, snow arrival and meltdown, and extreme winter warming events such as frost in the middle of winter due to snow cover absence (Table 1). Freezing days are the sum of days per study year where the mean daily temperature at ground level was below -1˚C, which indicates an absence of protective snow cover and hence vegetation is likely exposed to freezing temperatures. Precipitation data was gathered from publicly available maps and online databases [31, 32]. Soil pH was measured in blocks 1,6 and 10 at all sites during late August 2016 with the use of pH-indicator strips (Merck 109531 Color Card MColorpHast pH-indicator strips, pH 0–6).

## Vegetation sampling

Between 2009 and 2016, we recorded the frequency of vegetation presence of all vascular plant species in late August, by visually detecting the presence/absence of each species within 16 subplots ($0.0125 \, m^2$ each) per plot with a wooden frame of 50cm x 50cm (i.e. the frequency of occurrence of each species) [33]. Between 2017 and 2019, the point intercept method was applied to estimate vascular plant biomass where the abundance of vascular plant species was measured as the total number of hits on 20 pins within the plot [34]. All vascular plant species present within the plot but not hit were also accounted for giving them an abundance value of 0.1. Prior to statistical analyses, the point intercept estimates were converted to biomass estimates [35]. Pin ends touching bare soil, moss and lichen were also registered and represent a measure of their cover.

## Plant traits

To assess the impacts of vegetation recovery on ecosystem properties induced by the extreme event, we gathered information on three traits known to affect components of the carbon and/ or nitrogen cycles at the leaf, plant and ecosystem level [36, 37]. The species registered in all plots were assigned values for specific leaf area (SLA [$mm^2 \, mg^{-1}$]), leaf dry matter content (LDMC [$mg \, g^{-1}$]), and leaf nitrogen content (LNC [$mg \, g^{-1}$]) according to the LEDA [38] and TRY [39] databases. Trait values selected for each species were based on location, all species were assigned values belonging to northern Norway or Sweden. The aggregated values of leaf traits were calculated according to Garnier et al (2004), where the relative contribution of each species to the total biomass of the community was multiplied by the species trait value [36].

## Seed bank study

In the centre of all 144 plots, a seed bank sample was collected in spring 2009 using a 10-cm long metal cylinder (2.5 cm in diameter) placed down at soil level below the litter. The soil cores were kept in plastic bags and frozen at -18˚C within 8-h after collection. After 3 months, samples were thawed in dark chambers at 2˚C before they were crumbled onto petri dishes (8.7 cm in diameter) with a moist filter paper. All samples were placed in a phytotron chamber that simulated the optimal conditions for most northern alpine plants, that is, 72% relative humidity and 24h of simulated daylight (Philips TLD 489 fluorescent tubes) with 12h at 18˚C and 12h at 12˚C. All samples were sprayed with water every third day. Emerging seedlings were classified as dicotyledons or monocotyledons and removed from the petri dish. When no more seedlings emerged (about 12 weeks after the experiment was set up), all samples were given a new stratification period for 4 weeks in a dark chamber at 2˚C. After the stratification period, samples were placed in chambers with optimal growing conditions and seedling emergence was again registered for another 12 weeks.

## Seed rain study

Seed rain samples were collected annually from all five sites, between 2010 and 2016, by using one large green plastic doormat (FinnTurf™, 45x55 cm) per block (total of 10 doormats per site), stuck to the ground with 10 cm nails [40]. The mats were put out from early June and to mid-September to sample the current year seed rain. When collected, the mats were bagged and taken inside to dry at about 30 ˚C. Each mat was then turned up-side down and, with the help of a plastic hammer, the contents were removed and each sample was cleaned for *E. nigrum* leaves using a magnifying microscope, due to their phytotoxicity on seedling growth [16, 41]. The samples were then placed on petri dishes with a Whatman no. 4 filter paper for moisture and put in growing chambers with optimal conditions for seed germination. Emerging seedlings were identified as dicotyledon or monocotyledon and removed from the petri dish. After 12 weeks, the samples were given a stratification period in a dark chamber at 2˚C for four weeks. Thereafter, they were placed again in growing chambers with optimal conditions and emerging seedlings were again recorded for 12 more weeks.

## Seed addition study in the field

In autumn 2011, seeds of 7 vascular plant species were sown in all 144 plots, with help of the wooden frame used for vegetation sampling (16 subplots, 0.0125 m$^2$ each). Seeds were sown in 7 of the 16 subplots, with one species per subplot (Fig 1). The species sown were (1) *Avenella flexuosa* (ca.100 seeds), (2) *Solidago virgaurea* (ca.100 seeds), (3) *Calamagrostis phragmitoides* (ca.100 seeds), (4) *Chamerion angustifolium* (ca.100 seeds), (5) *Bistorta vivipara* (30 seeds), (6) *Rhinantus minor* (ca.50 seeds), and (7) *Melampyrun pratense* (5 seeds). Thus, we sowed a total of 485 seeds per plot and over 80.000 seeds across all sites. We chose faster growing species than *E. nigrum*, some that are known to grow together with *E. nigrum* (*A. flexuosa*, *S. virgaurea*, *B. vivipara* and *C. phragmitoides*) and some species that are not usually found in *E. nigrum* heathlands (*C. angustifolium*, *R. minor*, *M. pratense*). Seeds had been collected during autumn 2010 and stored in paper bags at 4 ˚C until the experiment took place. Seed germination viability was tested before they were sown out in the field and was for all species between 85% and 95%. Seedlings and established individuals were recorded each year at the end of August between 2011 and 2016.

## Bioassay study in the phytotron

Eight years after simulating the extreme event we evaluated the soil quality in the plots using seedling growth of four species as a bioassay. In autumn 2016, the top 5 cm of soil was sampled with a soil core (2.5cm diameter) from all plots. The soil cores were kept in plastic bags at -18˚C until the experiment took place. We chose half of the soil cores, from blocks 1,3,5,7 and 9, to have a representative sample per site. Soil was thawed and placed in plastic trays with 48 square pots of 5.5 cm x 5.5 cm. First, each pot was filled halfway with commercially available sand. Then the field collected soil was homogenized and put on top. We had 25 control samples where peat of commercially available quality was placed on top of the sand. Each hole was divided in three sections and three seedlings of *A. flexuosa*, *S. virgaurea* and *B. vivipara* were placed in each section. Seeds were germinated about one week before the experiment started and seedling growth was arrested in cold chambers at 2˚C once the root growth was 0.5 cm. The trays were placed in growth chambers under 24h of simulated daylight (Philips TLD 489 fluorescent tubes) with 12h at 18˚C and 12h at 12˚C. Seedlings were monitored every third day and water sprayed when necessary. The total shoot length of each seedling was measured after three weeks.

## Statistical analyses

All data were analysed using the statistical environment R [42]. Linear mixed effects models were used to account for our nested design of explanatory variables with a mixture of fixed and random effects [43]. The continentality index for each study site was used as a proxy for the effect of climate along the coast-inland gradient and was calculated according to Rivas-Martinez et al 2011, showing a range of 21.8 to 30.1 indicating a oceanic/maritime to continental climate [44]. The response to extreme event treatment with and without litter was similar for all variables, and data were pooled together for all analyses. We tested for main effects and interactions of the extreme event treatment and continentality index on species richness, mean biomass of all vascular plant species, mean *E. nigrum* biomass, and mean biomass of subordinate species in addition to changes in SLA, LDMC and LNC. Data from the bioassay and seed sowing experiment was also tested against the extreme event treatment and continentality index. Only data from 2019 were used for the statistical modelling since we were interested in studying the long-term effects of the treatment on the recovery of the vegetation along the climatic gradient. However, year was used as a predictor variable in the seed sowing experiment. Due to field work complications, data was missing from one of the sites (i.e. Snarby) for 2019 and therefore data from 2018 was used for this site. There are no statistical differences in the biomass between the last three years for any of the sites (F.value = 1.5, p = 0.93, Fig 2c) and we therefore chose to use data from 2018 for the missing site. No statistics were run on data from seed bank and seed rain due to the low germination rate.

## Results

A total of 21 vascular plant species were registered along the climatic gradient during the study (S1 Table), with a maximum of 7 vascular species per plot.

Control plots remained constant both in species richness and frequency during the 10 years (Fig 2). Moreover, all sites showed similar mean biomass of vascular plant species in control plots, as measured in 2019, with values between 275–350 g/m$^2$.

## Vegetation recovery

We encountered no new species during the study. All species were present in both extreme event and control plots, except for *Calluna vulgaris* which was not registered in extreme event

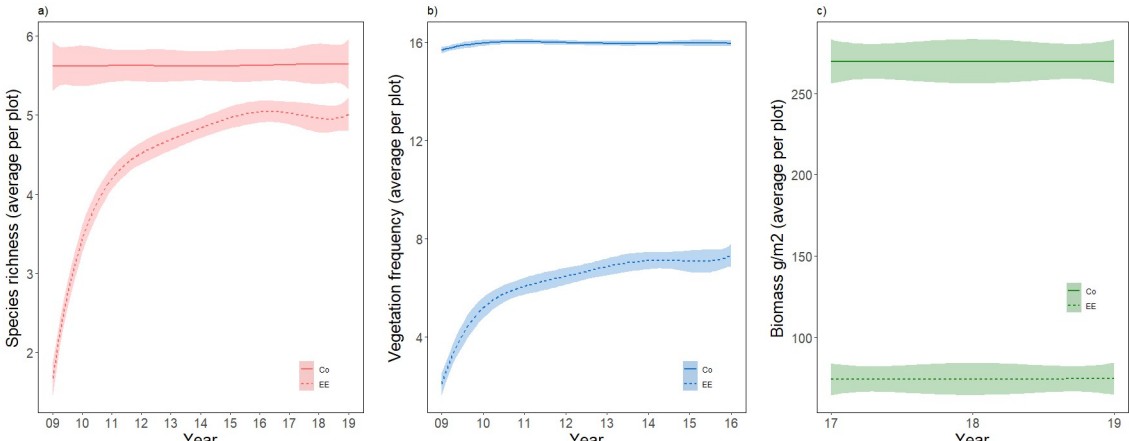

**Fig 2. a) Mean species richness (2009–2019), b) mean frequency of vegetation presence (2009–2016) and c) total biomass (g/m$^2$) (2017–2019) of all vascular plant species in control plots (Co) and extreme event (EE) plots registered throughout all study sites.** The mean is presented with 95% confidence intervals.

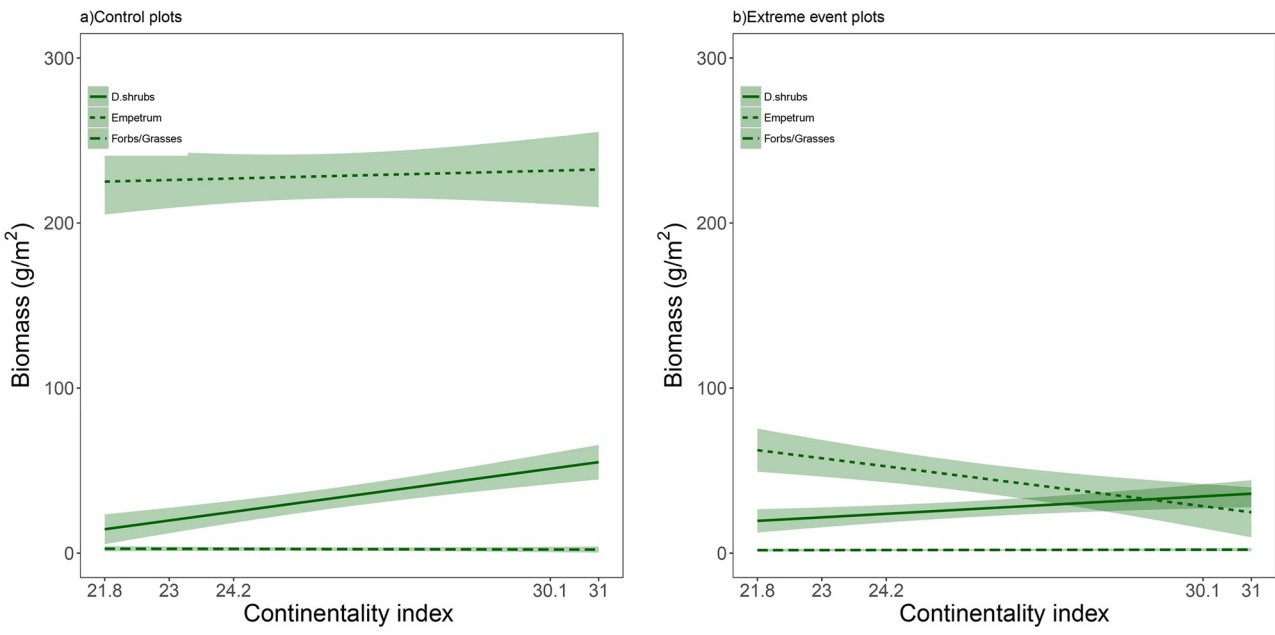

**Fig 3. Mean biomass (g/m$^2$) of dwarf shrubs (excluding *E. nigrum*) (i.e. D.shrubs), Forbs/grasses and only *E. nigrum* registered in 2019 along the climatic gradient (i.e. continentality index).** The mean is presented with 95% confidence intervals. a) Control plots, and b) extreme event plots.

plots. Furthermore, the speed of vegetation recovery was very low and close to stagnated about four years after the extreme event took place (Fig 2b). Thus, we found hardly any biomass in extreme event plots at the end of the study (Figs 3b and 4b). Extreme event plots showed also a slight decrease in total biomass with increasing continentality index, whereas control plots showed no such effect (Figs 3 and 5a, S2 Table).

Mean biomass of subordinate species alone (excluding *E. nigrum*) showed no difference between extreme event and control plots (Figs 3 and 5a, S2 Table). However, control plots had overall higher biomass of subordinate species in continental than coastal sites (Fig 5a), whereas

## a) Control vegetation          b) Vegetation nine years after extreme event

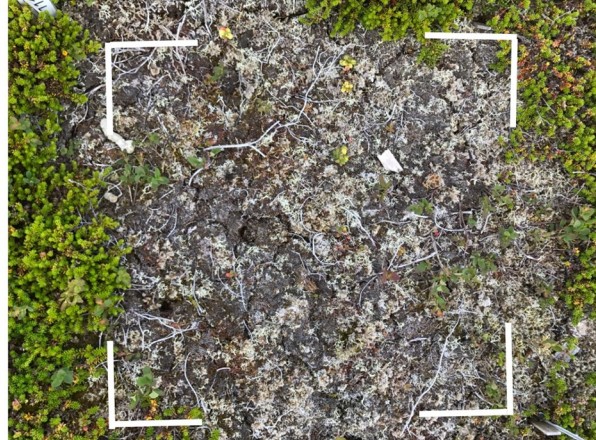

**Fig 4. Sample plot photographs taken during summer 2018.** a) Control plot, and b) Extreme event plot where the vegetation was removed 9 years ago. Take note of the low plant biomass and bare soil in the extreme event plot.

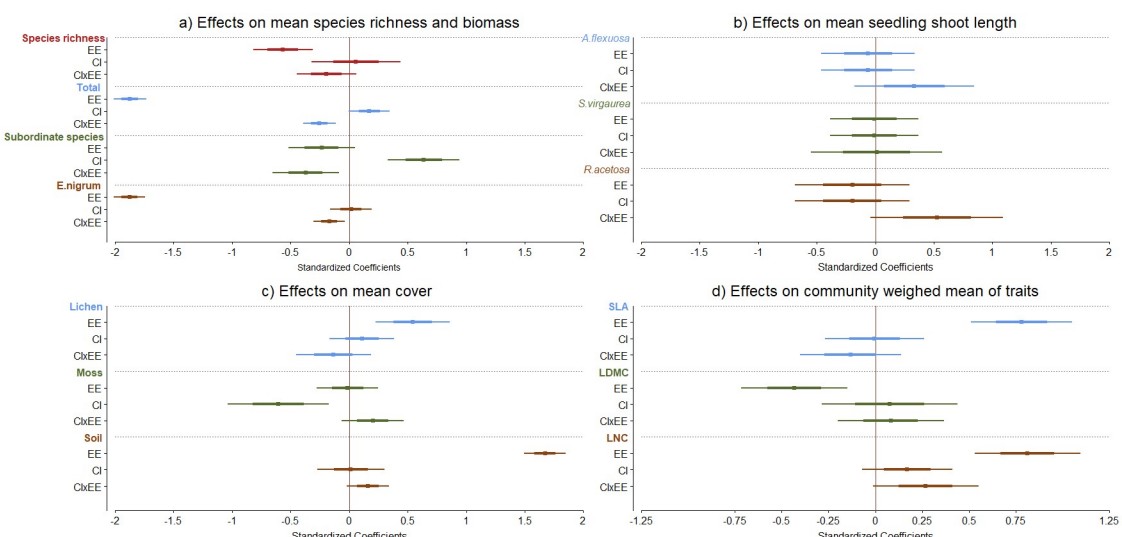

**Fig 5. Standardized coefficients from the linear mixed effects models on the effects of extreme event treatment (EE), continentality index (CI) and their interaction (CI x EE) on data collected in 2019.** a) mean species richness and biomass (total biomass, subordinate species biomass and *E. nigrum* biomass alone), b) mean seedling length of species used in bioassay experiment excluding peat treatment, c) mean cover of mosses and lichens, and bare soil, and d) community weighed means of selected traits, specific leaf area (SLA), leaf dry matter content (LDMC), leaf nitrogen content (LNC). Note the different scale of standardized coefficients in the different sections.

extreme event plots showed an opposite pattern, as seen from the significant interaction between extreme event treatment and continentality index (Fig 5a). Thus, continental sites had originally higher biomass of subordinate species, but also had lower vegetation recovery of these species after vegetation removal.

In contrast to subordinate species, we found a drastic reduction of mean *E. nigrum* biomass in extreme event plots when compared to control plots (Figs 3 and 5a). Also, we found a significant interaction of extreme event treatment and continentality index, with a higher biomass of *E. nigrum* in coastal than continental extreme event plots, but not in control plots. Thus, *E. nigrum* recovery after vegetation removal was promoted in coastal as compared to continental sites.

Bare soil and lichen cover were significantly higher in extreme event plots as compared to control plots, while moss cover varied overall across the climatic gradient, but not between extreme event and control plots, finding overall less moss cover in continental sites (Fig 5c, S2 Table). We found no difference in soil pH values between control plots and extreme event plots (F.value = 2.1, p = 0.92).

The vegetation trait composition nine years after the extreme event was simulated was characterized by higher values of both SLA and LNC, but lower values of LDMC in extreme event plots when compared to control plots (Fig 5d, S2 Table). These results were consistent along the climatic gradient, i.e. there was no significant interaction of extreme event treatment and continentality index (Fig 5d). In extreme event plots, mean SLA, LNC, and LDMC were 9.71 ±0.7mm, 14.18±0.5mg/g and 429.95 ± 3.21 mg/g respectively, while control plots had mean values of 6.88±0.5, 11.26±0.31, and 439.89 ±1.3 mg/g.

To further study the vegetation community changes after the extreme event, we grouped the species as dwarf shrubs or grass/forbs/sedges (S1 Table). We found that dwarf shrubs accounted for over 80% of the subordinate species biomass in control plots (Fig 3a) and that it was dwarf shrub biomass that made up most of the sparse vegetation regrowth in extreme

**Table 2. Germinable seeds m$^{-2}$ from the 144 seedbank samples and average seeds m$^{-2}$ per year from seed rain mats from each study site.**

|  |  | Study site | | | | |
|---|---|---|---|---|---|---|
|  |  | **Rebbenes** | **Skogsfjord** | **Snarby** | **Skibotn** | **Guhkesjávri** |
| Seed bank | Monocotyledon | 99.6 ± 13.7 | 0 | 0 | 24 ± 12.1 | 82 ± 65 |
|  | Dicotyledon | 60 ± 29.3 | 0 | 0 | 80 ± 2.5 | 56 ± 35.6 |
| Seed rain | Monocotyledon | 35.6 ± 10 | 9.2 ± 7 | 10.8 ± 2.5 | 7.2 ± 3 | 22.8 ± 12 |
|  | Dicotyledon | 20 ± 6 | 1.6 ± 0.5 | 27 ± 6 | 5 ± 1.4 | 13.1 ± 5 |

Seedlings were not identified to species but classified as monocotyledon or dicotyledon. The mean is presented with 95% confidence intervals.

event plots as well (Fig 3b). *E. nigrum* was the most important contributor to the dwarf shrub group in the three most coastal sites (Fig 3b). In the two most continental sites, *B. nana* and *E. nigrum* were equally abundant.

### Regenerative capacity of the *E. nigrum* heathlands

Seed germination from both the seed bank and seed rain was low for all sites (Table 2). We registered no germinating seeds from the seedbank samples in two of the study sites.

From the 2011 seed sowing experiment in the field, only three out of the seven species were registered as seedlings during the study (*A.flexuosa*, *S.virgaurea* and *B.vivipara*). Seedling counts were very low and decreased exponentially throughout the years (S1 Fig, S3 Table). On average, we registered a maximum of 5 seedlings per plot during the first year, and about one seedling the last year. The seedlings we registered over the years did not manage to establish into adult individuals. We found no difference between control plots and extreme event plots but found that continental sites had less registered germinating seedlings throughout the years (S1 Fig).

The bioassay in the phytotron showed that all four species had arrested growth when sown on soil collected from both extreme event and control plots when compared to soil from commercially available peat (S2 Fig). Furthermore, when comparing mean shoot length of all species in extreme event and control soil alone (excluding seedlings grown in peat), we found no effect of extreme event or continentality index except for a small effect on *S. virgaurea* (Fig 5b, S2 Table).

## Discussion

The results from our simulated extreme event on *E. nigrum* tundra heathlands along a sharp climatic gradient, shows extremely low vegetation recovery in all disturbed plots along the entire climatic gradient. Ten years after simulating the extreme event, the vegetation biomass and the species richness had not reached pre-disturbance levels and no new species had established in the disturbed plots. Bare soil and erosion were apparent in extreme event plots across the entire climatic gradient. Furthermore, our results showed a limited regenerative capacity of the *E. nigrum* heathlands in the form of both germinable seed bank and seed rain, and the addition to the plots of seeds of a range of faster growing vascular plant species did not contribute to a faster revegetation of the disturbed plots at any site. Overall, the consistency of the results along the climatic gradient confirms the strong legacy *E. nigrum* has on the soil environment and suggests that, the outcome of future extreme events on *E. nigrum* dominated heathlands could be mediated by the niche constructor ability of *E. nigrum* independent of the climatic conditions.

Our main hypothesis was that the niche constructor ability of *E. nigrum* would mediate any positive effects of climate on the recovery of *E. nigrum* heathlands after an extreme event.

Coastal sites did have a higher total biomass in extreme event plots. However, the biomass registered in extreme event plots was so small compared to control plots that the overall effect of climate had apparently negligible biological significance on the time scale of this study, thus confirming our hypothesis. Though plant trait values in extreme event plots differed from those in control plots, they were still typical of a slow growing late succession community [36] and belonged mostly to dwarf shrubs. A possible explanation for the near absence of grasses and forbs is that faster growing species, like *A. flexuosa* and *C. neglecta* were not present in high enough tiller density as to facilitate rapid regrowth [20] or that the low amount of germinable seeds in the seed bank and seed rain restricted the recruitment possibilities and hence vegetation recovery after disturbance [45]. However, the speed of recovery appeared to stagnate after four years and was not promoted by the addition of a large amount of seeds belonging to fast growing species, suggesting that it was not the lack of seed availability but other constrains to vegetation growth. Indeed, previous studies in continental *E. nigrum* heathlands where *E. nigrum* and/or neighbouring species were removed, found no change in plant community composition after seven [20] or eight years [21], which were attributed to the legacy effects left by *E. nigrum* in the soil environment, mainly through the production and release of phytochemicals [46]. The results presented here are in accordance with these studies, but further confirm that niche construction by *E. nigrum* is not only apparent in continental areas but also across an entire gradient of climatic conditions ranging from oceanic to continental climate. Thus, because *E. nigrum* is able to dominate heathlands under a range of habitats and climates [13, 23], a potential increase of extreme events not tolerated by *E. nigrum* could bring lasting changes to large areas of the subarctic tundra dominated by *E. nigrum*. Our results also suggest that under an increase in extreme events scenario, *E. nigrum* and potentially other slow growing ecosystem retarding species will gradually regain dominance over the disturbed areas. This is substantiated by that *E. nigrum* has a positive growth response even after being exposed to significant cold spells in winter [9], suggesting it can tolerate a range of extreme events and still keep, and potentially extend, its dominance.

We found higher lichen cover in extreme event plots than in control plots, suggesting that extreme events could contribute to an increased lichen biomass in tundra areas previously dominated by evergreen shrubs [47, 48]. Nevertheless, bare soil was much more common than cover of lichen, moss and vegetation biomass (Fig 4), indicating that large sections of the plots were bare, what could contribute to erosion and soil degradation over time. Thus an increase in extreme events on *E. nigrum* heathlands could potentially promote the browning trend which is being registered in some parts of the tundra [49].

Overall, we show how a common niche constructor species can dramatically affect ecosystem recovery after extreme events independent of climate being continental or oceanic. Our study contributes to the growing literature on the important ecosystem-modifying role of *E. nigrum* in tundra plant communities.

## Supporting information

**S1 Fig. Mean seedling count from seed sowing experiment per site (± 95% confidence intervals) between 2012–2016 across all sites.** Continentality index per site is presented in parenthesis. Species included *Avenella flexuosa*, *Solidago virgaurea*, *Calamagrostis phragmitoides*, *Chamerion angustifolium*, *Bistorta vivipara*, *Rhinantus minor* and *Melampyrun pratense*. (DOCX)

**S2 Fig. Mean shoot length from bioassay experiment (± 95% confidence intervals).** Species used were *Avenella flexuosa*, *Rumex acetosa* and *Solidago virgaurea* growing for three weeks in each of the three soil types (i.e. sampled from control plots (C), from extreme event plots (EE)

in 2016, and commercially available peat (Peat)).
(DOCX)

**S1 Table. List of vascular plant species and growth forms registered in extreme event and control plots in 2019 in order of decreasing biomass (g/m$^2$).**
(DOCX)

**S2 Table. Anova tables from the linear models showing the effect of the extreme event (EE), continentality index (CI) and their interactions (CIxEE) on the mean values of a) species richness, total biomass, biomass of subordinate species and biomass of *E. nigrum* alone; b) leaf traits: specific leaf area (SLA), leaf dry matter content (LDMC) and leaf nitrogen content (LNC), c) seedling length of the vascular plant species used in bioassay, and d) lichen, moss and soil.** Significant values are presented in bold.
(DOCX)

**S3 Table. ANOVA table from the linear model showing the effect of the extreme event (EE), continentality index (CI), year, and the interaction of year and continentality index (YearxCI) on the mean seedling count of all species sown in 2011.** Only significant interactions are presented. Significant values are presented in bold.
(DOCX)

## Acknowledgments

We are thankful to Sissel Kaino, Xavier Ancin, Anna-Katharina Pilsbacher, Metha Klock, Mildrid Svoen and Mikel Moriana Armendariz for help during the field work, and Leidulf Lund for help performing the studies at the phytotron.

## Author Contributions

**Conceptualization:** Victoria T. González, Bente Lindgård, Rigmor Reiersen, Kari Anne Bråthen.

**Data curation:** Victoria T. González, Bente Lindgård, Rigmor Reiersen, Snorre B. Hagen, Kari Anne Bråthen.

**Formal analysis:** Victoria T. González, Snorre B. Hagen.

**Investigation:** Victoria T. González, Kari Anne Bråthen.

**Methodology:** Victoria T. González, Rigmor Reiersen, Kari Anne Bråthen.

**Project administration:** Kari Anne Bråthen.

**Supervision:** Kari Anne Bråthen.

**Validation:** Victoria T. González, Kari Anne Bråthen.

**Visualization:** Victoria T. González.

**Writing – original draft:** Victoria T. González.

**Writing – review & editing:** Victoria T. González, Bente Lindgård, Rigmor Reiersen, Snorre B. Hagen, Kari Anne Bråthen.

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
