## [Decision Letter · Decision Letter 0]

2 Dec 2020

PONE-D-20-33120

NICHE CONSTRUCTION MEDIATES CLIMATE EFFECTS ON RECOVERY OF TUNDRA HEATHLANDS AFTER EXTREME EVENT

PLOS ONE

Dear Dr. Gonzalez,

Thank you for submitting your manuscript to PLOS ONE. After careful consideration, we feel that it has merit but does not fully meet PLOS ONE’s publication criteria as it currently stands. Therefore, we invite you to submit a revised version of the manuscript that addresses the points raised during the review process.

Please make sure that all data is publicly available.

We look forward to receiving your revised manuscript.

Kind regards,

Craig Eliot Coleman, PhD

Academic Editor

PLOS ONE

Journal Requirements:

"Unfunded studies"

5. We note that Figure 1 in your submission contains map images which may be copyrighted. All PLOS content is published under the Creative Commons Attribution License (CC BY 4.0), which means that the manuscript, images, and Supporting Information files will be freely available online, and any third party is permitted to access, download, copy, distribute, and use these materials in any way, even commercially, with proper attribution. For these reasons, we cannot publish previously copyrighted maps or satellite images created using proprietary data, such as Google software (Google Maps, Street View, and Earth). For more information, see our copyright guidelines: http://journals.plos.org/plosone/s/licenses-and-copyright.

(1) You may seek permission from the original copyright holder of Figure 1 to publish the content specifically under the CC BY 4.0 license. 

Reviewers' comments:

Reviewer's Responses to Questions

**Comments to the Author**

1. Is the manuscript technically sound, and do the data support the conclusions?

Reviewer #1: Yes

Reviewer #2: Yes

2. Has the statistical analysis been performed appropriately and rigorously? 

Reviewer #1: Yes

Reviewer #2: Yes

3. Have the authors made all data underlying the findings in their manuscript fully available?

Reviewer #1: No

Reviewer #2: Yes

4. Is the manuscript presented in an intelligible fashion and written in standard English?

Reviewer #1: Yes

Reviewer #2: Yes

5. Review Comments to the Author

Reviewer #1: The study by Gonzales et al. presents the results of a very interesting long-term experiment on how Empetrum nigrum heaths recover from extreme events, such as pathogens or winter warming events. The authors show that this species recover very slowly, with little biomass returning after ten years, and that the presence of E. nigrum also prevents the establishment of other species. This is an important indication that areas dominated by E. nigrum may be harder hit in the long-term by recurring extreme events than areas dominated by other species. I have very few comments on this paper, which I think is well-executed and fills an important knowledge gap on the long-term impact of extreme events on these ecosystems.

Some minor comments:

Line 103 and line 106: This is confusing. Were the study sites situated in the tree line or just above the tree line?

line 158: 'mean daily temperature': is this soil temperature? Please specify

Line 241: 'were' should be 'where'

Lines 55, 67, 94, 149 and 374: 'further' should be 'furthermore'

Reviewer #2: This study aims to assess the role played by a niche constructor species after an extreme event (i.e., removing aboveground vegetation) along long environmental gradient and over ten years in tundra heatlands in Northern Norway. This is a very interesting and well-conducted study, that certainly will attract the attention of PLOS ONES's readers. Studies exploring vegetation recovery in such a long gradient over a long period of time are scarce even though they are extremely needed. Although in general I'm very sympathetic with the story, I miss some explanations for a couple of issues.

First, I miss a more thorough development for plant traits extraction from LEDA and TRY databases. Please, add more details about how trait values were assigned. Did you look for trait values of your species in the same location that you are sampling? Or did you extracted trait values from different locations?

Regarding aggregated values of leaf traits, did you calculate them for each year?

Finally, B. nana, another dwarf species, and E. nigrum were equally abundant in the two most continental sites (L340). I was wondering if in those sites won´t be expected that B. nana will have an important role in vegetation recovery as they could also act as niche constructors. Is there any work with this species that shows that it also has that role?

Minor comments

L168 I think that you don’t need to repeat that you are recording the frequency of all plants as it is in L165.

L168-172 “Vegetation frequency appeared to stabilize…”

It is not clear when plant biomass was estimated, please clarify. The first part of the sentence looks like a result.

L172 “All vascular plant species present within the plot but not hit were also accounted for giving them an abundance value of 0.1”

I’m not very familiar with the method, thus I was wondering if this is a common procedure when using the point intercept method? I though that you just consider plants that hit the pin.

L219 Correct: one species

L255 “We chose faster growing species than E. Nigrum”

Please add which ones within brackets.

6. PLOS authors have the option to publish the peer review history of their article (what does this mean?). If published, this will include your full peer review and any attached files.

Reviewer #1: **Yes: **Frans-Jan Parmentier

Reviewer #2: No

---

## [Author Response · Author response to Decision Letter 0]

8 Jan 2021

Response to comments from academic editor: 

Answer: Manuscript has been revised and adjusted to the PLOS ONE guidelines. 

Answer: Study sites were located in areas with little to no easy access to the public on state owned land and therefore no permits were needed to establish and follow up the study. This has been incorporated in the methods section

 Answer: Data has been uploaded to BIRD database and is available on the following link. https://hdl.handle.net/11250/2722169

"Unfunded studies"

a. Please clarify the sources of funding (financial or material support) for your study. List the grants or organizations that supported your study, including funding received from your institution. Answer: The study was funded by a grant from the Arctic University of Norway provided to Victoria T. Gonzalez.

b. State what role the funders took in the study. If the funders had no role in your study, please state: “The funders had no role in study design, data collection and analysis, decision to publish, or preparation of the manuscript.” Answer: The funders had no role in study design, data collection and analysis, decision to publish, or preparation of the manuscript.

d. If you did not receive any funding for this study, please state: “The authors received no specific funding for this work.”

5. We note that Figure 1 in your submission contains map images which may be copyrighted. All PLOS content is published under the Creative Commons Attribution License (CC BY 4.0), which means that the manuscript, images, and Supporting Information files will be freely available online, and any third party is permitted to access, download, copy, distribute, and use these materials in any way, even commercially, with proper attribution. For these reasons, we cannot publish previously copyrighted maps or satellite images created using proprietary data, such as Google software (Google Maps, Street View, and Earth). For more information, see our copyright guidelines: http://journals.plos.org/plosone/s/licenses-and-copyright.

We require you to either (1) present written permission from the copyright holder to publish these figures specifically under the CC BY 4.0 license, or (2) remove the figures from your submission.

 Answer: The map figure provided in Figure 1 was obtained from the platform R, the free software environment for statistical computing and graphics. Maps and code used in the program R are not subject to any copyright and can be freely distributed. 

Answer: Captions have been included at the end of the manuscript and in-text has been changed. 

Response to reviewer comments:

Reviewer #1: The study by Gonzales et al. presents the results of a very interesting long-term experiment on how Empetrum nigrum heaths recover from extreme events, such as pathogens or winter warming events. The authors show that this species recover very slowly, with little biomass returning after ten years, and that the presence of E. nigrum also prevents the establishment of other species. This is an important indication that areas dominated by E. nigrum may be harder hit in the long-term by recurring extreme events than areas dominated by other species. I have very few comments on this paper, which I think is well-executed and fills an important knowledge gap on the long-term impact of extreme events on these ecosystems. 

Answer: Thank you for this positive review of our study. 

Some minor comments:

Line 103 and line 106: This is confusing. Were the study sites situated in the tree line or just above the tree line? 

Answer: Study sites were situated above the tree line, corrected in text.

line 158: 'mean daily temperature': is this soil temperature? Please specify.

 Answer: We refer to the mean daily temperature at ground level where the temperature loggers were placed. This has been specified in the text.

Line 241: 'were' should be 'where'.

 Answer: Spelling has been corrected in the text

Lines 55, 67, 94, 149 and 374: 'further' should be 'furthermore'. 

Answer: Spelling has been corrected in the text

Reviewer #2: This study aims to assess the role played by a niche constructor species after an extreme event (i.e., removing aboveground vegetation) along long environmental gradient and over ten years in tundra heatlands in Northern Norway. This is a very interesting and well-conducted study, that certainly will attract the attention of PLOS ONES's readers. Studies exploring vegetation recovery in such a long gradient over a long period of time are scarce even though they are extremely needed. Although in general I'm very sympathetic with the story, I miss some explanations for a couple of issues. 

Answer: Thank you for this positive review of our study.

First, I miss a more thorough development for plant traits extraction from LEDA and TRY databases. Please, add more details about how trait values were assigned. Did you look for trait values of your species in the same location that you are sampling? Or did you extracted trait values from different locations? 

Answer: The plant traits values were extracted from The LEDA and TRY databases based on their location. Those values belonging to locations close to our field studies were chosen. All species have values belonging to individuals in northern Norway or northern Sweden. This has now been included in the methods section. 

Regarding aggregated values of leaf traits, did you calculate them for each year?

 Answer: Only the last year of data was used in the statistical analyses (data from 2019). This is stated in the “Statistical Analyses” section

Finally, B. nana, another dwarf species, and E. nigrum were equally abundant in the two most continental sites (L340). I was wondering if in those sites won´t be expected that B. nana will have an important role in vegetation recovery as they could also act as niche constructors. Is there any work with this species that shows that it also has that role? 

Answer: It has been shown in northern areas that biodiversity increases with shrub canopy height of among other species Betula nana (Bråthen and Lortie, “A portfolio effect of shrub canopy height on species richness in both stressful and competitive environments”, Functional ecology 2016, 30,60-69). The result of an increase of B. nana could bring an increase in biodiversity on the long run, however, the biomass of B.nana and other dwarf shrubs after ten years of study is extremely low and does not really allow us to say anything conclusive about their effect. 

Minor comments

L168 I think that you don’t need to repeat that you are recording the frequency of all plants as it is in L165. 

Answer: Sentence has been removed

L168-172 “Vegetation frequency appeared to stabilize…”

It is not clear when plant biomass was estimated, please clarify. The first part of the sentence looks like a result. 

Answer: Plant biomass was estimated for each year between 2017 and 2019. Sentence has been removed as suggested to avoid confusion.

L172 “All vascular plant species present within the plot but not hit were also accounted for giving them an abundance value of 0.1”

I’m not very familiar with the method, thus I was wondering if this is a common procedure when using the point intercept method? I though that you just consider plants that hit the pin. 

Answer: In this method it is a common procedure to give those species not touched by the pins a “symbolic” value in order for the data to have an overview of all species present and so be able to calculate other parameters such as species richness or biodiversity. 

L219 Correct: one species Answer: text has been corrected

L255 “We chose faster growing species than E. Nigrum”

Please add which ones within brackets. 

Answer: the species are already listed on the same sentence (line 250-253 of revised manuscript), [quote]: “We chose faster growing species than E. nigrum, some that are known to grow together with E. nigrum (A. flexuosa, S. virgaurea, B. vivipara and C. phragmitodes) and some species that are not usually found in E. nigrum heathlands (C. angustifolium, R. minor, M. pratense)”

---

## [Editor Report · Decision Letter 1]

11 Jan 2021

NICHE CONSTRUCTION MEDIATES CLIMATE EFFECTS ON RECOVERY OF TUNDRA HEATHLANDS AFTER EXTREME EVENT

PONE-D-20-33120R1

Dear Dr. Gonzalez,

We’re pleased to inform you that your manuscript has been judged scientifically suitable for publication and will be formally accepted for publication once it meets all outstanding technical requirements.

Kind regards,

Craig Eliot Coleman, PhD

Academic Editor

PLOS ONE
---

## [Editor Report · Acceptance letter]

14 Jan 2021

PONE-D-20-33120R1 

Niche construction mediates climate effects on recovery of tundra heathlands after extreme event 

Dear Dr. Gonzalez:

I'm pleased to inform you that your manuscript has been deemed suitable for publication in PLOS ONE. Congratulations! Your manuscript is now with our production department. 

Kind regards, 

on behalf of

Dr. Craig Eliot Coleman 

Academic Editor

PLOS ONE